# The Impact of Heatwaves on Mortality and Morbidity and the Associated Vulnerability Factors: A Systematic Review

**DOI:** 10.3390/ijerph192316356

**Published:** 2022-12-06

**Authors:** Fadly Syah Arsad, Rozita Hod, Norfazilah Ahmad, Rohaida Ismail, Norlen Mohamed, Mazni Baharom, Yelmizaitun Osman, Mohd Firdaus Mohd Radi, Fredolin Tangang

**Affiliations:** 1Department of Community Health, Faculty of Medicine, Universiti Kebangsaan Malaysia, Bandar Tun Razak, Kuala Lumpur 56000, Malaysia; 2Environmental Health Research Centre, Institute for Medical Research, Ministry of Health Malaysia, Shah Alam 40170, Malaysia; 3Environmental Health Unit, Disease Control Division, Ministry of Health Malaysia, Putrajaya 62590, Malaysia; 4Occupational and Environmental Health Unit, Kelantan State Health Department, Ministry of Health Malaysia, Kota Bharu 15590, Malaysia; 5Surveillance Unit, Kedah State Health Department, Ministry of Health Malaysia, Alor Setar 05400, Malaysia; 6Department of Earth Sciences and Environment, Faculty of Science and Technology, Universiti Kebangsaan Malaysia, Bangi 43600, Malaysia

**Keywords:** heatwave, mortality, morbidity, vulnerability, sensitivity, systematic review

## Abstract

Background: This study aims to investigate the current impacts of extreme temperature and heatwaves on human health in terms of both mortality and morbidity. This systematic review analyzed the impact of heatwaves on mortality, morbidity, and the associated vulnerability factors, focusing on the sensitivity component. Methods: This systematic review was conducted according to the Preferred Reporting Items for Systematic Reviews and Meta-Analyses 2020 flow checklist. Four databases (Scopus, Web of Science, EBSCOhost, PubMed) were searched for articles published from 2012 to 2022. Those eligible were evaluated using the Navigation Guide Systematic Review framework. Results: A total of 32 articles were included in the systematic review. Heatwave events increased mortality and morbidity incidence. Sociodemographic (elderly, children, male, female, low socioeconomic, low education), medical conditions (cardiopulmonary diseases, renal disease, diabetes, mental disease), and rural areas were crucial vulnerability factors. Conclusions: While mortality and morbidity are critical aspects for measuring the impact of heatwaves on human health, the sensitivity in the context of sociodemographic, medical conditions, and locality posed a higher vulnerability to certain groups. Therefore, further research on climate change and health impacts on vulnerability may help stakeholders strategize effective plans to reduce the effect of heatwaves.

## 1. Introduction

The anthropogenic greenhouse has impacted the earth, resulting in climate change and increasing the global temperature. This is referred to as global warming and has caused extreme climate and weather events such as heatwaves, heavy rainfall, and drought [1,2,3]. The latest Intergovernmental Panel on Climate Change (IPCC) report showed that heatwaves have become more common and intense over the past 50 years [4].

The impacts of heatwaves on human health are evident, with health outcomes from mortality and morbidity reported. A heatwave event can be significant and often disastrous, as shown by excess deaths during the European heatwave of 2003 and the Central European and Russian heat wave of 2010 [5,6,7].

Given the devastating impact of extreme heat and heatwave events, many researchers have attempted to understand the vulnerability factors in the population [8,9,10,11]. Human vulnerability to climate change or variability is a complicated concept and has no universally accepted definition [12]. The study of the vulnerability of humans to climate change is one of the key concepts to understanding their ability to adapt to changes in climate hazards [13]. Approaches for assessing vulnerability have progressed since previous IPCC assessments. For example, the Fourth Assessment Report (AR4) of the IPCC expressed that vulnerability is a function of three factors, which are exposure, sensitivity, and adaptive capacity [14]. The AR5 of the IPCC proposed the concept of risk. Risk is the function of four factors, which are hazard (new term in AR5), exposure, sensitivity, and adaptive capacity. Since then, studies have been conducted to examine the link between the new concept of risk in AR5 and the existing concept of vulnerability in AR4 [15,16,17]. The IPCC has published its AR6 and defined vulnerability as “the propensity or predisposition to be adversely affected and encompasses a variety of concepts and elements, including sensitivity or susceptibility to harm and lack of capacity to cope and adapt” [4]. These frameworks were formulated for all climate change hazards. 

Despite the progress in the science of vulnerability, there is no systematic and consistent conceptual framework specifically for heat hazard [18]. There is a growing body of literature on heat vulnerability frameworks focusing on various components such as exposure, sensitivity, and adaptive capacity [18,19,20,21]. Heat vulnerability is how likely a person is to be injured or harmed during hot weather. Extreme heat, such as heatwaves, can impair thermoregulation and affect multiple organs, including the heart, lungs, renal system, central nervous system, and digestive tract [22]. Impaired thermoregulation, for example, can cause dehydration, increased blood viscosity, and burden the heart function, ultimately leading to cardiac failure [23]. Additionally, heatwave occurrences impact mental illness [24]. 

Some studies conceptualized heat vulnerability as comprising two elements: human exposure to heat and human sensitivity [25]. Wilhelmi and Hayden (2010) proposed a heat vulnerability framework and described that the sensitivity component to heat involves an individual’s characteristics (health status, socio-demographics, etc.) as well as certain aspects of the community where one lives (environment, community demographics) [26]. These factors can play an essential role in one’s ability to adapt to heat.

Several studies have determined that heatwave events will severely affect vulnerable groups such as the elderly, infants, and people with pre-existing chronic conditions [27,28,29]. However, some inconsistent results were shown in other studies [11,30,31], influenced by the variability in the mortality and morbidity indicators, statistical analysis methods used, and differences in the population’s sensitivity, adaptive capacity, and coping mechanisms. Another factor causing these inconsistencies was the variability in heatwave definitions and locality factors. To date, there is no standardized definition of a heatwave. As an example, some studies used a daily temperature exceeding 90th to 98th percentile for at least two consecutive days [31,32,33]. Meanwhile, a study conducted in Thailand used 30 different heatwave definitions [34]. These heterogeneous definitions made it challenging to estimate the pooling of effect estimation of heatwave impact on human health. In addition, locality factors (continents and geographical conditions) may significantly influence the temperature of each locality [4].

Furthermore, most of the literature involving health impact studies was confined to mortality or morbidity rather than actual health outcomes [35,36,37,38]. Thus, this review aims to present heatwaves’ impact on mortality and morbidity and identify vulnerability, focusing on the sensitivity components for both outcomes.

## 2. Materials and Methods

This study, guided by the PRISMA (Preferred Reporting Items for Systematic Reviews and Meta-Analyses) review protocol, was explicitly designed for systematic reviews and meta-analyses [39], following the systematic review protocol [40] registered on PROSPERO (CRD42021232847).

### 2.1. Inclusion Criteria

It was formulated based on PECO [41], a tool based on three main concepts: participants (human), exposure (heatwave), comparator (mortality and morbidity), and outcomes (vulnerability factors). Included were research-based articles, peer-reviewed, English publications from 2012 to 2022 focused on heatwaves, study findings comprising both heatwave-related mortality/morbidity impact and vulnerability factors (focusing on the sensitivity component), and involved human participants.

### 2.2. Study Selection and Data Search

Related articles were identified by searching the Scopus, Web of Science, EBSCO host, and PubMed databases. The search string was created and generated using Boolean operators and keyword search (Appendix A). Following the removal of duplicates, two reviewers independently examined the titles and abstracts of all identified studies to select the articles based on the predetermined selection criteria.

### 2.3. Quality Assessment

The article’s quality was determined by using the Navigation Guide Systematic Review framework [41] (Appendix A). The third reviewer resolved any disagreement.

### 2.4. Data Extraction and Synthesis

Following the initial search, we created a standardized form to extract the following data: author and study year, study design, study location, type of climate, meteorological data, heatwave definitions, health data, statistical analysis, heatwave impact on mortality and morbidity, as well as the sensitivity component of vulnerability assessment.

Selected articles were divided into two groups based on mortality and morbidity outcomes. Sensitivity factors and statistical results from the quantitative analysis were thoroughly described using a narrative synthesis. Finally, the findings of the selected articles were merged using a narrative approach for the overall results.

## 3. Results

A total of 32 articles were selected and analyzed to identify the impact of heatwaves on mortality, morbidity, and the associated vulnerability factors (focusing on the sensitivity component) (Figure 1). Most articles involved time series studies (*n* = 27), and the rest were case-crossover studies (*n =* 5). The included articles represented most of the continents in the world: Asia, 14; Oceania, 8; Europe, 5; North America, 4; and South America, 1. In addition, the articles spanned lower, upper, middle, and high-income countries. Most of the studies were conducted in warm and temperate regions (*n =* 19) and cold and temperate (*n =* 7). The rest of the studies were conducted in the tropical region (*n =* 3), low and subarctic region (*n =* 2), and temperate region (*n =* 1). Fifteen articles described mortality’s impacts and vulnerability factors (focusing on the sensitivity component), while another eighteen addressed morbidity. Table 1 summarises the characteristics and main findings of the studies included in this systematic review.

### 3.1. Mortality

#### 3.1.1. Heatwave Impact on Mortality

The included studies had various mortality indicators. The main mortality indicators were overall (all-cause) mortality, non-external cause of mortality, and cause-specific mortality. Most articles (*n =* 14) reported a significant association between heatwaves and mortality.

For the overall (all-cause) mortality rate, several articles showed a statistically significant increment during heatwave exposure. A study by Cheng et al. (2018) showed an increased overall mortality rate of 28% (95% CI: 15–42%) [49]. Meanwhile, in Korea, overall mortality risk increased during heatwaves by 11.6% (95%): 7.8–15.5%) [59]. A study in Iran showed that deaths from non-external causes increased significantly during heatwaves (RR = 1.03, 95% CI: 1.01, 1.05; adjusted ozone: RR = 1.09, 95% CI: 1.07, 1.09); PM 10 (particulate matter ≤ 10 µm wide) adjusted: RR = 1.09, 95% CI: 1.07, 1.09) [27].

Meanwhile, some studies reported significant effects of heatwaves on cause-specific mortality. As an example, a study in Finland reported significant related cardiovascular mortality (PI 9.9%, 95% CI 7.7–12.1%) [62]. Another study in China reported increased cardiopulmonary-related mortality during heatwaves (RR 1.07, 95% CI: 1.03, 1.10) [50].

#### 3.1.2. Sensitivity Component of Vulnerability Assessment for Mortality

The highest-ranked sensitivity of the vulnerability assessment was elderly age, with eleven articles. Here, we studied two categories of the elderly, ages > 65 years and >75 years. A study in China showed that people aged >65 years comprised a high percentage of the total non-accidental mortality data (average annual loss (AAL) = 61.3%, 95% CI: 30.6, 91.9) [33], while another by Wang et al. (2015) reported that the >75-year age group had the highest mortality (RR 1.46, 95% CI: 1.28, 1.66) [43]. The female and male gender had the second and third highest reported sensitivity of the vulnerability assessment with five and four articles, respectively. A study conducted in China reported that the female gender had a higher risk of heatwave-related cardiovascular mortality compared to males [50]. Another study in Finland reported that the female gender had a higher heatwave-related non-accidental mortality risk with 12.5% (95% CI: 9.1–16.0%) [62]. Meanwhile, a study in Australia reported that the male gender had a 1.22 times higher risk for heatwave-related non-accidental cause mortality (RR 1.22, 95% CI: 1.05, 1.42) [43]. A study in a cold temperate region also showed that the heatwave-related non-accidental mortality risk increased by 7.2% (95% CI: 3.3–12.0%) among the male gender [62]. Three articles reported that people with cardiovascular diseases had a significant mortality risk due to heatwave exposure. The study in Tehran showed that cardiovascular disease contributed about 52% of the total cause of death [27]. Two articles reported that people with respiratory diseases showed significant mortality risk due to heatwave exposure. People with respiratory disease accounted for 52% total cause of heatwave-related mortality [27].

The other essential sensitivities of the vulnerability assessment were low education level, renal disease, mental disease, diabetes, and rural area, with one article each. Table 2 and Table 3 show additional information.

### 3.2. Morbidity

#### 3.2.1. Heatwave Impact on Morbidity

Eighteen articles showed a significant association between heatwave exposure for the morbidity impact. In this systematic review, morbidity was classified into heat-related illness (heatstroke), hospital admission (non-specific), cardiovascular-related hospital admission (non-specific, arrhythmia), respiratory-related hospital admission (non-specific, asthma, chronic obstructive pulmonary disease), infectious-related admission, urinary-related admission, Alzheimer’s disease-related admission, diabetes-related hospitalizations, emergency department (ED) visit, and ambulance callout. A USA study reported that excess respiratory admissions due to heatwaves would be 2 to 6 times higher from 2080 to 2099 than in 1991–2004 [42]. Meanwhile, a study in South Korea showed that heatwaves increased cardiovascular-related hospital admission by 14% [45]. Significant increment of heatwave-related urinary disease admissions by 88.3% compared to non-heatwave days [48]. Another in Australia reported an effect on hospitalization for diabetes during heatwaves (OR 1.37, 95% CI: 1.11, 1.69) [55].

#### 3.2.2. Sensitivity Component of the Vulnerability Assessment for Morbidity

The highest-ranked sensitivity of the vulnerability assessment was being elderly, with 13 articles reporting similar findings. A study in Australia that measured the impact of heatwaves on hospital ED visits showed that patients aged >75 years had significant risk factors (RR = 1.28, 95% CI: 1.09, 1.50) [11]. A study in the USA showed that the elderly had a significant association with hyperthermia-related hospitalization (RR 11.4, 95% CI: 9.55, 13.25) [56].

Seven articles reported that children had a significant association with heatwave-related morbidity. Heatwaves increased the risk of all-cause hospitalizations among children by 11% [31]. For heatwave-related ambulance callouts, children < 5 years old have a significant sensitivity in the vulnerability assessment (OR 1.36, 95% CI: 1.10, 1.68) [61]. The male gender followed this with six articles. A study in the USA showed male gender had a significant risk for heatwave-related asthma hospitalization (Odds ratio (OR) 1.12, 95% CI: 1.04, 1.22) [9] and a significant vulnerability assessment for cardiovascular-related admission by Kang et al. (2016) [45]. A study on heatwave-related ambulance callouts showed male gender as a significant sensitivity of the vulnerability assessment (RR 1.03, 95% CI: 1.02, 1.03) [54].

Heatwaves significantly impacted patients with cardiorespiratory diseases such as asthma, chronic obstructive pulmonary diseases, and pneumonia, found in this study [9,11,42,46]. In addition, a study in Australia found that respiratory and cardiovascular diseases increased emergency department visits by 2% and 1%, respectively, during heatwave days [11]. A population-based retrospective cohort study showed a significant increase in hospitalization for diabetic patients during heatwaves [55]. Five articles reported low socioeconomic status was significantly associated with heatwave-related morbidity. A study by Toloo et al. (2014) showed that low socioeconomic status was associated with increased emergency department visits by 12% compared to non-heatwave days [11]. Meanwhile, four articles reported female gender is a significant risk factor for heatwave-related morbidity. Another study in Vietnam showed that the female gender was associated with increased all-cause hospitalizations by 8.1% (95% CI: 2.6–13.9%) [46]. Table 2 and Table 3 show additional information.

## 4. Discussion

### 4.1. Heatwave Impact on Mortality and Morbidity

Included articles in this systematic review showed that exposure to heatwaves negatively impacts mortality and morbidity. However, the impact varied across studies and regions. For example, the literature included in the present study had multifactorial causes of mortality and morbidity indicators. Most of the causes of death were related to the cardiovascular and respiratory systems, which might be due to these diseases being the most common worldwide. The impact on morbidity also had varying underlying causes. However, the most common cause of heatwave-related hospitalizations was similar to that of mortality; cardiopulmonary-related diseases.

### 4.2. Sensitivity Component of Vulnerability Assessment

This current review adopted the sensitivity component of the vulnerability assessment proposed by a previous study [26]. It is ideal for studying all the components of vulnerability assessment for a more comprehensive finding. However, the diversity of vulnerability conceptualizations is seen in different contexts, referring to different systems being exposed to different hazards [13,66,67]. In this review, three major sensitivities of the vulnerability assessment factors were identified: sociodemographic, medical conditions, and locality characteristics.

#### 4.2.1. Sociodemographic

Age was one of the critical factors for the population’s vulnerability to heatwave exposure. The review shows that the impact of heatwaves on mortality and morbidity involves all age groups, the elderly group being the most vulnerable, explained by the decreased efficiency in body temperature regulation because of aging [68]. In addition, it is common for this group to live with co-morbid chronic medical illnesses such as heart disease and chronic obstructive pulmonary disease (COPD) [69]. Thus, combined risk factors of age and co-morbidity increase the susceptibility to heat-related illnesses. Additionally, children are at risk due to their underdeveloped body regulatory systems [70]. They are also vulnerable as they tend to spend more time playing outdoors [70]. People in the working-age group are also at risk from the impact of heatwaves due to their work activities [71]. 

In this review, the risk for heatwave impacts varies across gender, which can be influenced by factors such as pre-existing medical conditions, social support, and the type of work exposure of a particular person. For example, males may be at risk in some locations where outdoor work roles are predominantly male-dominated [28]. However, females may be at greater risk as women had a higher risk due to a high surface-to-mass ratio and greater subcutaneous fat thickness than men [72]. Low socioeconomic status has been identified as a common risk factor for any disease or health issues, including vulnerability to the impact of heatwaves [73,74]. There are reasons that make those living in low socioeconomic classes more vulnerable to the effects of heatwaves. Lower socioeconomic status is commonly associated with other medical conditions, such as malnutrition and infectious disease, which can aggravate the impact of heatwaves on health. In addition, because of their financial status, this group has less access to household amenities, such as fans, proper housing ventilation, and AC, which are essential for reducing the impact of heatwaves by cooling the body temperature. The nature of jobs commonly associated with outdoor work, such as construction and menial jobs, exposes people to high temperatures. Using fans and air conditioners reduces the risk of heat-related death [75]. These amenities are essential for faster body cooling and avoiding the harmful effects of heatwaves. This will be a problem, particularly for people in low and middle-income countries. 

A lack of knowledge of heatwaves may expose a person to the negative impact of heatwaves on health. Most media sources provide vital information on heatwaves, such as their effects and preventive measures. However, better understanding requires knowledge and education levels. Thus, a person with a lower education level is subject to a more significant impact than a more educated person.

#### 4.2.2. Medical Conditions

People with specific medical conditions are vulnerable to heatwaves. In the present systematic review, several medical conditions showed significant evidence of heatwave adverse effects on mortality and morbidity.

People with cardiovascular disease are affected by heatwave exposure. The underlying pathophysiologic mechanism for the relation between heat stress and cardiovascular disease, such as increased red and white cell counts in the circulation, leads to increased blood viscosity, platelet release into the bloodstream, and reduced plasma volume [76]. 

A similar impact is seen in people with respiratory disease. Human thermal regulation attempts to maintain a safe body temperature during heatwave exposure, resulting in increased cardiac output and hyperventilation. Consequently, the respiratory rate and tidal volume will increase, worsening respiratory diseases such as asthma and COPD and requiring hospitalization [77]. People with renal disease can be compromised during a heatwave. The underlying mechanism is that exposure to heatwaves increases dehydration risk and leads to electrolyte imbalance [78], which imposes extra stress on renal function and exacerbates pre-existing renal diseases. Autonomic neuropathy in diabetic patients makes them more vulnerable to heatwave effects [79]. 

In addition, heatwaves increase the risk of mental health-related outcomes. One possible explanation is exposing people to psychological trauma associated with higher frequency, intensity, and duration of climate-related disasters, including extreme heat exposure or heatwave events [80]. Identifying the population suffering from these medical conditions could help local health authorities and service providers incorporate mental health impacts into their heatwave warning systems, as well as develop public health policies and guidelines to address preventable heat-related mental health mortality and morbidity.

However, the findings of these medical conditions on mortality and morbidity varied between studies. This scenario can be explained by the variability of other factors influencing the outcome. Thus, further studies are warranted to address this uncertainty.

#### 4.2.3. Locality

In rural areas, the population is at risk of the impact of heatwaves, possibly due to socioeconomic factors, medical infrastructure, and the aging population. For example, economic activities in rural areas are conducted outdoors, and a heatwave’s impact during extreme heat can be amplified. Thus, evaluating this population’s knowledge, perception, and adaptive behavior is crucial for an early preventive plan.

Meanwhile, urban areas are commonly associated with the urban heat island (UHI) phenomenon. The UHI effect is mainly due to human activities and construction that lead to heat accumulation [81]. The effects of UHI can be mitigated by improved energy efficiency, urban landscape optimization, green roof construction, high reflectivity material utilization, and green land cultivation. Different climatic zones also play a role in determining the population’s heat sensitivity. Different climatic zones had different effects in this review. Acclimatization of the people, behavioral adaptations, medical infrastructure, availability of heat warning systems, and other factors could play a role.

Most of the literature included in this review was from developed countries. One possible explanation is that these countries have sufficient resources for studying this topic and publishing their findings. Developing and warmer countries, such as Southeast Asia, experience more frequent, long-lasting, and intense heatwaves [82]. However, there are limited publications from these countries, which may underestimate the burden of heatwaves on these particular countries.

## 5. Conclusions

Mortality and morbidity indicators, including all-cause and cause-specific, are critical for measuring heatwave impacts on human health. The sensitivity in the context of sociodemographic, medical conditions, and locality posed a higher heat vulnerability to certain groups. The impact of heatwaves on mortality and morbidity involves all age groups, especially the elderly and children. People with specific medical conditions, particularly cardiovascular and respiratory diseases, are most vulnerable to heatwave impacts of mortality and morbidity. The impact of heatwaves on mortality and morbidity and their associated vulnerability factors varies depending on the locality. These findings can help stakeholders strategize effective plans to reduce the effects of heatwaves according to their target populations and respective areas. Nevertheless, further study on the other component of vulnerability assessment, such as adaptive capacity, will provide more information on identifying vulnerable populations.

## Figures and Tables

**Figure 1 ijerph-19-16356-f001:**
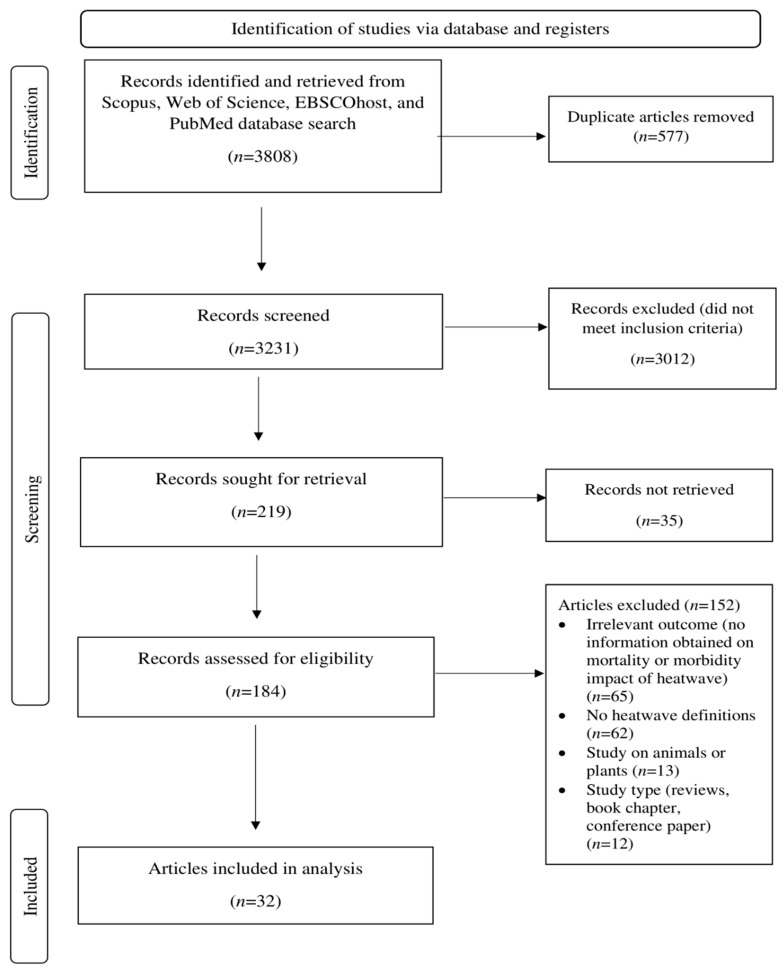
The PRISMA flow diagram.

**Table 1 ijerph-19-16356-t001:** Characteristics and main findings of the selected studies (*n =* 30).

Author Year	Study Design	Study Region (Country)	Type of Climate	Meteorological Data	Heatwave Definitions	Health Data	Statistical Analysis	Impact	Sensitivity Component of Vulnerability Assessment
Lin et al., 2012 [42]	Time series	New York (USA)	Warm and temperate	Hourly temperature, barometric pressure, dew point, ozone	90th percentile of apparent temperature (AT) based on the summer AT distribution from 1991–2004	Respiratory admissions	GAM	Excess respiratory admissions due to extreme heat/heatwave would be 2 to 6 times higher in 2080–2099 than in 1991–2004	Female (1.35% higher risk) Age > 75 (1.17% higher risk) Low income (1.26% higher risk)
Ahmadnezhad et al., 2013 [27]	Time series	Tehran (Iran)	Warm and temperate	Daily maximum temperature, daily mean temperature, daily minimum temperature, air pollutants (ozone, PM_2.5_, PM_10_)	Maximum temperature above 90th percentile for three consecutive days	Mortality data - Non-external cause - Cause-specific (cardiovascular, cerebrovascular, respiratory)	GLLM	Non-external cause of death increases significantly during heatwaves RR 1.03, 95% CI: 1.01, 1.05 (adjusted ozone) RR 1.09, 95% CI: 1.07, 1.09 (adjusted PM_10_)	Age > 65 years old (18.2% of total excess death) Female (1.05 times higher than male) Cardiovascular disease (52% of total cause of death, *p* = 0.001) Respiratory disease (33.4% of total death, *p* = 0.02)
Toloo et al., 2014 [11]	Time series	Brisbane (Australia)	Warm and temperate	Daily maximum temperature, daily mean temperature, daily minimum temperature, air pollutants (ozone, PM_10_)	Daily mean temperature above the 95th percentile for two or more consecutive days	Emergency department (ED) presentations	GAM	Respiratory presentations increased 2% during heatwaves Cardiovascular disease presentations increased 1% during heatwaves	Male (RR 1.10, 95% CI: 1.02, 1.09) Age > 75 years old (RR 1.28, 95% CI: 1.09, 1.50) Low socioeconomic (ED presentation increased 12% compared to non-heatwave days)
Wang et al., 2015 [43]	Time series	Brisbane, Melbourne, and Sydney (Australia)	Warm and temperate	Daily maximum temperature, daily minimum temperature, relative humidity	Mean temperature above a certain percentile (90th, 95th, 98th, 99th) for two or more consecutive days	Mortality data - Non-accidental - Cause-specific (circulatory)	GAM	Significant heatwave-related non-accidental mortality—highest during summer season.RR 1.40 (95% CI: 1.26, 1.55)	Female (RR 1.56, 95% CI: 1.36, 1.79) Age > 75 years old (RR 1.46, 95% CI: 1.28, 1.66)
Tong et al., 2015 [29]	Time series	Brisbane, Melbourne, and Sydney (Australia)	Warm and temperate	Daily maximum temperature, daily mean temperature, daily minimum temperature, relative humidity	Daily mean temperature above 75th⋯99th percentiles for 2 or more consecutive days	Mortality data - Non-accidental	Poisson-GAM	Significant increase in mortality during heatwave Highest RR 1.34 (95% CI: 1.22, 1.46)	Female (RR 1.52, 95% CI: 1.35, 1.71) Elderly (RR 1.54, 95% CI: 1.34, 1.77)
Green et al., 2016 [44]	Time series	(United Kingdom)	Warm and temperate	Mean Central England Temperature (CET)	Mean CET > 20 °C at least three consecutive days	Mortality data - All-cause	Linear regression model	No significant heatwave-related excess mortality	Elderly (102 deaths per heatwave day, 95% CI: 88–115)
Soneja et al., 2016 [9]	Time-stratified case-crossover	Maryland (USA)	Cold and temperate	Daily maximum temperature, total precipitation	Daily maximum temperature above 95th percentile	Asthma hospitalizations	Conditional logistic regression	Heatwave-asthma hospitalization OR 1.23, CI:1.15, 1.33	Male (OR 1.12 95% CI: 1.04, 1.22) Age < 4 years old (OR 1.20 95% CI: 1.05, 1.37)
Kang et al., 2016 [45]	Time series	(South Korea)	Cold and temperate	Daily mean temperature, relative humidity, air pressure, air pollutants (CO, ozone, NO_2_, SO_3_, PM_10_)	Daily mean temperature above the 98th percentile for at least two consecutive days	Cardiovascular related hospitalization	GAM Conditional logistic regression	Heatwave significantly associated with cardiovascular-related hospital admission (14% increased admissions)	Male and elderly aged ≥ 65 years *p* = 0.039
Phung et al., 2017 [46]	Time series	(Vietnam)	Tropical	Daily maximum temperature, daily mean temperature, daily minimum temperature, relative humidity, cumulative rainfall	A measure of apparent temperature ≥ 90th percentile for the 3 preceding days or more for the summer in northern cities and for the whole year in southern cities	Hospitalizations - All-cause - Cardiovascular disease - Respiratory disease - Infectious disease	GAM DLM GLM	Heatwave event was associated with increased hospital admissions: - All causes (7.1–12.7%) - Cardiovascular diseases (6.8–31.3%) - Infectious diseases (9.8–27.3%) - Respiratory diseases (2.8–23.2%)	Female (RR 8.1%, 95% CI: 2.6–13.9) *
Xu et al., 2017 [32]	Time series analysis	Brisbane (Australia)	Warm and temperate	Daily maximum temperature, daily mean temperature, daily minimum temperature, relative humidity	Daily mean temperature at 90th, 95th, and 97th percentile of the temperature distribution for 2, 3, or 4 days	Hospitalization - All-cause	Poisson-GAM DLNM	Significant heatwave-related hospitalization RR > 1	Children
Li et al., 2017 [47]	Time series	Chongqing (China)	Warm and temperate	Daily maximum temperature, daily mean temperature, daily minimum, daily mean relative humidity	≥3 consecutive days with daily average temperature equal to or over the threshold temperature	Heatstroke-related hospitalizations	Zero-inflated Poisson regression model (ZIP) with a logistic distribution	90.2% of heatstroke cases occurred during heatwaves	Elderly (>65 years old) (Highest excess risk (ER) 32.3% on lag2)
Borg et al., 2018 [48]	Time series	Adelaide (Australia)	Warm and temperate	Daily maximum bulb temperature, daily minimum bulb temperature	Daily calculation of the Excess Heat Factor (EHF) index	Admissions for urinary diseases	Negative binomial (NB) regression models	Significant heatwave-related urinary diseases admissions (88.3% increase in ED admissions compared to non-heatwave days) IRRs 1.883, 95% CI 1.531–2.315	Male has higher ED presentations for total urinary diseases (IRRs > 1) Age > 65 years old has higher ED presentation for total urinary diseases (IRRs > 1)
Cheng et al., 2018 [49]	Time series analysis	(Australia)	Warm and temperate	Daily maximum temperature, daily mean temperature, daily minimum temperature	Daily mean temperature above certain percentile (95th to 99th) of the temperature distribution that lasts for several days in the warm season (November to March of next year).	Mortality data among elderly - All-cause	Quasi-Poisson regression Random effect meta-analysis	Significant heatwave-mortality average death increased 28% (95% CI: 15–42%)	Elderly (28% increased mortality risk)
Yin et al., 2018 [50]	Time series	(China)	Tropical and subarctic	Daily mean temperature, daily mean relative humidity, air pollutants (ozone, PM_10_)	Daily mean temperature above certain percentile (90th, 92.5th, 95th to 97.5th) of the temperature distribution that lasts for 2, 3, and 4 days	Mortality data - Non-accidental cause - Cause-specific (cardiovascular, coronary heart disease, strokes, respiratory disease, chronic obstructive pulmonary disease)	GAM	Heatwave-related total cause mortality RR 1.07, 95% CI: 1.03, 1.10	Elderly (RR > 1) Female (RR > 1)
Huang et al., 2018 [34]	Time series	(Thailand)	Tropical	Daily mean temperature, relative humidity	30 heatwave definitions used 10 intensities (90th, 91st, 92nd, …, or 99th percentile of the mean temperature across the study period) and three durations (i.e., ≥2, 3, or 4 consecutive days) ^†^	Mortality data - Cause-specific (infectious diseases, neoplasms, endocrine, metabolic diseases, diabetes mellitus, circulatory system, ischemic heart disease, respiratory system, pneumonia, digestive system, genitourinary, renal)	Quasi-Poisson GAM Random effects meta-analysis DLNM Meta-regression analysis	Heatwave associated with increased on: - Non-external cause mortality RR 1.126, 95% CI: 1.103, 1.150 - Ischemic heart disease RR 1.219, 95% CI: 1.134, 1.311 - Pneumonia RR 1.184, 95% CI: 1.104, 1.269	Elderly ^#^ Lower education ^#^
Zhang et al., 2018 [33]	Time series	(China)	Tropical and subarctic	Daily maximum temperature, daily mean temperature, daily minimum, relative humidity	Daily average temperature > 98th percentile for >2 consecutive days or Daily maximum temperature > 35 °C for >2 consecutive days or Daily maximum temperature > 95th percentile for >2 consecutive days	Mortality data - Non-accidental	DLNM, Monte Carlo analysis	Heatwave-predicted AAL during 2051–2095 will increase 8–90 times compared the ALL during 1971–2015	Age > 65 years old (AAL, 61.3, 95% CI: 30.6, 91.9)
Campbell et al., 2019 [51]	Case cross-over	Tasmania (Australia)	Warm and temperate	Daily maximum temperature, daily mean temperature, daily minimum, air pollutants	Daily mean temperature (DMT) averaged over the three-day period (TDP) is higher than the climatological 95th percentile for DMT	Emergency department admissions	Conditional multivariate logistic regression	Heatwave-related ED presentation increased by 5% - (OR 1.05, 95% CI 1.01, 1.09)	Children ≤ 14 years old (OR 1.13, 95% CI 1.03,1.24)
Li et al., 2019 [52]	Time series	Shelby County (USA)	Warm and temperate	Daily maximum temperature, daily mean temperature, daily minimum	Maximum daily temperature > 95th percentile for more than two consecutive days	Mortality data - All-cause - Cause-specific (respirator, hyperthermia, circulatory)	Poisson regression models DLNM	Significant heatwave-related cardiovascular mortality RR: 1.25, 95% CI: 1.01, 1.55	No significant effect by socioeconomic, race, or urbanicity
Xu et al., 2019 [53]	Case cross-over	Brisbane (Australia)	Warm and temperate	Daily maximum temperature, daily mean temperature, daily minimum temperature, relative humidity, air pollutants (NO_2_, PM_10_)	Daily mean temperature above 90th, 95th, and 97th percentiles for 2 consecutive days	Hospitalizations for Alzheimer’s disease	Conditional logistic regression	Intense heatwaves increased the risk of hospitalizations for Alzheimer’s disease (*n =* 907) Odds ratio (OR) > 1	Female (51.9% higher risk) Elderly (>65 years old) contributed 93.3% of hospitalizations
Patel et al., 2019 [54]	Time series	Perth (Australia)	Warm and temperate	Daily temperature, air pollutants (CO, SO_2_, NO_2_, ozone, PM_10/2.5_)	Excess Heat Factor (EHF) value > 0	Ambulance callout	Single and multiple risk factor analyses Poisson regression modeling	Significant heatwave-related ambulance callout RR 1.10, 95% CI: 1.08, 1.12	Male (RR 1.03, 95% CI: 1.02, 1.03) Elderly (>60 years old) (RR 1.01, 95% CI: 1.00, 1.01) Low to middle socioeconomic index area (RR 1.02, 95% CI: 1.02, 1.03)
Zhao et al., 2019 [31]	Time series	(Brazil)	Tropical	Daily maximum temperature, daily mean temperature, daily minimum, relative humidity	12 heatwave definitions (Combining thresholds at the 90th, 92.5th, 95th, or 97.5th percentiles of city-specific year round daily mean temperatures and durations of 2, 3, or 4 consecutive days)	hospitalizations - All-cause	Quasi-Poisson regression DLM Random-effect meta-analysis	Heatwave increased risk of hospitalization, 26%, (95% CI: 1.9%, 3.2%)	Children (0–9 years old)- 11% higher risk of hospitalizations Elderly (age > 70 years old)- 18% higher risk of hospitalizations
Xu et al., 2019 [55]	Case cross-over	Brisbane (Australia)	Warm and temperate	Maximum temperature, minimum temperature, relative humidity, air pollutants	Daily mean temperature > 90th percentile for two or more days	Diabetes-related hospitalizations and mortality data with diabetes as the primary cause of death	Conditional logistic regression, case-only design with bi-nary/multinomial logistic regression	Significant effect on hospitalization for diabetes during heatwaves - OR 1.19, 95% CI: 1.02, 1.39 (95th percentile) - OR 1.37, 95% CI: 1.11, 1.69 (97th percentile) Significant effect on post-discharge death due to diabetes during heatwaves - OR 1.46, 95% CI 1.03, 2.07 (90th percentile) - OR 2.37, 95% CI 1.39, 4.03 (97th percentile)	Hospitalizations among children (0–14 years old) - OR 1.51, 95% CI 1.08, 2.60 (97th per-centile) - OR 1.49, 95% CI 1.01, 2.20 (95th per-centile) - OR 1.36, 95% CI 1.04, 1.78 (90th per-centile)
Liss et al., 2019 [56]	Time series	(USA)	Temperate	Daily maximum temperature, daily minimum	Any day when the nighttime temperature is above 90th percentile for the current and previous nights	Hyperthermia-related hospitalizations among elderly	Harmonic negative binomial generalized linear model (HNBGLM) with the log-link function	Highest RR hyperthermia-related hospitalization during heatwaves RR 11.4, 95% CI: 9.55, 13.25	Elderly (RR 11.4, 95% CI: 9.55, 13.25)
Patel et al., 2019 [57]	Time series	Perth (Australia)	Warm and temperate	Daily mean temperature, air pollution	Excess Heat Factor (EHF) > 0	Daily emergency department admissions (EDA)	Poisson regression modeling	Emergency department admission (EDA) rate was higher on heatwave days compared with non-heatwave days Rate Ratio (RR): 1.053, 95% CI 1.048, 1.058	- Elderly (>60 years old), RR 1.04, 95% CI 1.039, 1.049 - Low socioeconomic status, RR 1.083, 95% CI 1.079, 1.087
Kim et al., 2020 [58]	Time series	(South Korea)	Cold and temperate	Daily mean temperature	Daily mean temperature above the 95th percentile of the temperature distribution for two or more consecutive days	Mortality data (elderly population) - All-cause	Pearson’s correlation GLM with quasi-Poisson distribution DLM	Heatwave-mortality risk Percent Increase (PI) 11.6%, 95% CI: 7.8–15.5%	Elderly female: (PI: 14.7%, 95% CI: 9.2–20.4%) Elderly male: (PI: 6.9%, CI: 1.7–12.4%) Covariate: social isolation
Kang et al., 2020 [59]	Two-stage time series	(South Korea)	Cold and temperate	Daily mean temperature	Daily mean temperatures above certain percentiles (85th to 99th percentile) of the summer temperature distribution for >2 days	Mortality data - All-cause - Cause-specific (cardiorespiratory, non-cardiorespiratory)	GLM with quasi-Poisson distribution with a DLM	Significant heatwave-mortality (all-cause) risk RR 1.11, 95% CI: 1.01, 1.22	Rural (RR: 1.23, 95% CI: 0.99, 1.53) Elderly > 65 years old (RR 1.13, 95% CI: 1.05, 1.21)
Sohail et al., 2020 [60]	Time series	Helsinki (Finland)	Cold and temperate	Daily mean temperature, air pollutants	(a) Daily mean temperature above 90th percentile for four or more consecutive days (b) Daily mean temperature above 95th percentile for three or more consecutive days	Non-elective hospital admissions (cardiovascular disease, all respiratory disease, cerebrovascular disease, arrhythmia, asthma, chronic obstructive pulmonary disease (COPD), pneumonia)	Poisson regression-GLM	Heatwave-related pneumonia admissions associated with 25% increased risk (95% CI: 6.9%, 35.9%)	Majority (46.3%) of all cardiorespiratory hospital admissions occurred among persons aged >75 years
Campbell et al., 2021 [61]	Case cross-over	Tasmania (Australia)	Warm and temperate	Daily mean temperature, air pollutants	Daily calculation of the Excess Heat Factor (EHF) index	Ambulance dispatches	Conditional multivariate logistic regression	Significant heatwave-related ambulance dispatches Extreme heatwave: OR 1.34 (95% CI: 1.18, 1.52) Severe heatwave: OR 1.10 (95% CI: 1.05, 1.15) Low heatwave: OR 1.04 (95% CI: 1.02, 1.06)	Similar risk between male and female (OR > 1) Children <5 years old (OR 1.36), (95% CI: 1.10, 1.68) Elderly > 65 years old: (OR 1.47, 95% CI: 1.21, 1.78) Low socioeconomic: (OR 1.40, 95% CI: 1.18, 1.65)
Kollanus et al., 2021 [62]	Time series	(Finland)	Cold and temperate	Daily mean temperature	Daily mean temperature exceeded the 90th percentile for 4 or more days	Mortality data - Non-accidental	GEE	During heatwaves, non-accidental mortality risk increased by 9.9%, 95% CI: 7.7%, 12.1%	Age > 65 years old (12.8%, 95% CI: 9.8–15.9%) Women (12.5%, 95% CI 9.1–16.0%) Men (7.2%, 95% CI 4.4–10.0%) Cardiovascular disease (97.6%, 95% CI: 3.3–12.0%) Respiratory disease (25.3%, 95% CI: 16.0–35.3%) Renal disease (38.4%, 95% CI: 12.5–70.3%) Mental disorder (29.7%, 95% CI: 21.3–38.6%)
Wondmagegn et al., 2021 [63]	Time series	Adelaide (Australia)	Warm and temperate	Daily mean temperature	Excess Heat Factor (EHF)	Emergency department visits	DLNM	ED presentations (all-cause) were generally higher during heatwave days relative to non-heatwave days, 1162, 95% CI: 342, 1944	Age > 65 years old 554, 95% CI: 228, 834 Age 0–14 years 449, 95% CI: 173, 702
Thompson et al., 2022 [64]	Time series	(England)	Warm and temperate	Daily maximum temperature, daily mean temperature, daily minimum temperature	Mean CET > 20 °C at least three consecutive days	Mortality data - All-cause	Episode analysis Poisson distribution	Total estimate of the all-cause excess mortality during heatwave events: 1807 (95% CI 1575 to 2037)	Elderly (>65 years old) constitute 85% of total number of mortalities
Graczyk et al., 2022 [65]	Time series	(Poland)	Cold and temperate	Daily maximum temperature, daily minimum	At least 3 consecutive days with a daily maximum temperature above 30 °C	Mortality data - Non-external	Student *t*-test DLNM	Heatwave-related natural cause mortality risk increased by 20–146%	Number of natural cause mortality was 87% higher than expected among elderly population

Abbreviations: AT = apparent temperature; ALL = average annual loss; CET = Central England Temperature; CI = confidence interval; CO = carbon monoxide; DLNM = distributed lag non-linear model; DLM = distributed lag model; ED = emergency department; EHF = excess heat factor; ER = excess risk; GAM = generalized additive model; GEE = generalized estimating equation; GLLM = generalized linear lag model; GLM = generalized linear model; HNBGLM = harmonic negative-binomial generalized linear model; IRRs = incidence rate ratio/s; NO_2_ = nitric oxide; OR = odds ratio; PI = percent increase; PM = particulate matter; RR = relative risk; SO_3_ = sulfur trioxide; USA = United States of America. * All-cause hospitalizations; ^#^ Based on Cochrane test and I2 statistic; ^†^ Example.

**Table 2 ijerph-19-16356-t002:** Summary of sensitivity component of vulnerability assessment for heat-related mortality.

Sensitivity Component of Vulnerability Assessment			Article/s with Significant Association (*n*)
Sociodemographic	Gender	Female	*n =* 5 [27,43,50,58,62]
		Male	*n =* 4 [28,29,43,58,62]
	Age	Elderly	*n =* 11 [27,33,34,43,44,49,50,59,62,64,65]
	Low education		*n =* 1 [34]
Medical conditions	Cardiovascular disease		*n =* 3 [27,28,62]
	Respiratory disease		*n =* 2 [27,62]
	Renal disease		*n =* 1 [62]
	Mental disease		*n =* 1 [62]
	Diabetes		*n =* 1 [55]
Locality	Rural		*n =* 1 [59]

**Table 3 ijerph-19-16356-t003:** Summary of sensitivity component of vulnerability assessment for heat-related morbidity.

Sensitivity Component of Vulnerability Assessment			Article/s with Significant Association (*n*)
Sociodemographic	Gender	Female	*n =* 4 [42,46,53,61]
		Male	*n =* 6 [9,11,45,48,54,61]
	Age	Elderly	*n =* 13 [11,31,42,45,47,48,53,54,56,57,60,61,63]
		Children	*n =* 7 [9,31,32,51,55,61,63]
	Low socioeconomic		*n =* 5 [11,42,54,57,61]
Medical conditions	Respiratory disease		*n =* 4 [9,11,42,46]
	Cardiovascular disease		*n =* 3 [11,45,46]
	Diabetes		*n =* 1 [55]

## Data Availability

Data are contained within the article or Appendix A.

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
