# Peer review of "The Impact of Heatwaves on Mortality and Morbidity and the Associated Vulnerability Factors: A Systematic Review"

_ijerph, 2022, doi:10.3390/ijerph192316356_

Round 1

Reviewer 1 Report (Previous Reviewer 2)

Thank you for considering the previous comments. The following minor points remain outstanding.

Line 44 and 71: The full name of the IPCC remains incorrect. The correct term is ‘Intergovernmental Panel on Climate Change’, not ‘International Panel on Climate Change’.

Line 232: Table 2 and 3, not Table 1 and 3

Line 362: “The impact of heat waves impact on mortality and morbidity…” poor language, please reword.

Author Response

Reviewer 2 Report (Previous Reviewer 3)

Dear Authors, 

Thank you very much for revising the manuscript. The revisions have addressed several of the previous comments. It is also good to see that the authors got paper proofread by a professional service. Generally, the paper has been much strengthened now. However, I am still confused as in why vulnerability and sensitivity are mixed up. Vulnerability and sensitivity are two different terms although they are a function of each other. This is why I have previously directed the authors to IPCC vulnerability framework. However. I have not seen rebuttal related to that comment on standard IPCC vulnerability framework. The conclusions section is too small. It is very awkward to see a heatwave definition at the end of the paper (pg 22.); please take it to the literature review. The limitations could go into conclusions.

Author Response

This manuscript is a resubmission of an earlier submission. The following is a list of the peer review reports and author responses from that submission.

Round 1

Reviewer 1 Report

Comments attached in the document.

Author Response

Dear Reviewer,

Thank you

Reviewer 2 Report

Thank you for the opportunity to review the manuscript “The impact of heatwaves on mortality and morbidity and the associated vulnerability factors: A systematic review”.

While the premise of this manuscript is sound, there is some methodological and editorial reworking needed.

Initially, the manuscript needs a full English language edit throughout. It is difficult to read in places with multiple grammatical errors which significantly detract from the understanding of the paper. I have not made specific reference to these errors in my comments below.

Introduction

The IPCC is variously described as the ‘Intergovernmental Commission on Climate Change’ or the ‘International Governmental Panel of Climate’. Please reword to the correct name throughout.

The introduction would benefit from the inclusion of an additional paragraph outlining the biomedical pathways of heat illness on various organ systems. This would help justify the use of specific conditions in the search strings used in the review.

I am not clear as to the gap this paper is attempting to address. What is the specific issue that needs clarification from this research?

Line 60: Include the abbreviation ‘AC’ where used for the first time, as this abbreviation is used later in the paper.

Methods

Including ‘global warming’, ‘climate change’ or ‘greenhouse effect’ in the search string does not match with the intention of the review, which is specifically regarding heatwaves. Similarly, the omission of ‘ambulance*’ or ‘GP visit*’ or ‘primary care’ or ‘emergency presentation*’ or ‘emergency department*’ does not cover the breadth of mortality outcomes from heatwave events. Please undertake the review again with these changes. I would also recommend adding ‘heat*’ to cover the term ‘heat wave’ which is commonly used (as opposed to ‘heatwave’).

Various chronic conditions (other than cardiovascular, respiratory and renal disease) are known to be affected by extreme heat. Please review the literature and include addition disease states.

Line 105-6: “The statistical outcomes from the quantitative analysis are thoroughly described.” Where? Please provide a location in the paper.

Results

Line 118-120: Australia is a separate continent to Asia. As this sentence reads, there appears to be no papers from Australia, however papers from Australia are listed in Table 2. Please revise.

Line 179: As hospital emergency department visits and ambulance callouts were not included in the search initial string, it is unlikely you have the full picture here. This will need to be reviewed with the revised search string.

Discussion

Line 212-213: ‘Most of the causes of death were related to the cardiovascular and respiratory systems’. As these were the conditions that were used in the search string, its likely to see these are the primary diseases. This can be reviewed once other conditions are used in the revised search string.

Lines 235-237: These appear as gendered assumptions rather than evidence-based reasonings. Please provide references for these statements.

Lines 238-239: Please provide a reference for the statement on low SES and health.

Lines 273-274: This is a previously recognised outcome of heatwaves. Please review the literature on this and include this condition in your search string. Then revise this section according to the results.

Lines 293-99: This statement does not correlate with your results. Only 7 of the 35 studies were from European countries. Please revise this paragraph to accurately reflect your results. You may want to refer to Campbell et al. 2018 for a greater understanding of heatwave research and global location.

Conclusion

Lines 324-326: In Section 4.4, you state that ‘each locality setting must be examined for a better understanding’, which is a thoughtful and accurate response based on the differences in your findings in different locations. However, this statement is not correlated in the conclusion, where you state policymakers can use these findings to reduce the impact of heatwaves. This statement implies a generalisability that is not present in your findings. It would be helpful to include a statement on generalisability of your findings in the discussion, and then draw on this in your conclusion.

Author Response

Dear Reviewer,

Thank you

Reviewer 3 Report

Thank you very much for giving an opportunity to review this paper.

The paper reviews impact of heatwaves on mortality and morbidity, and the associated vulnerability using a systematic review methodology, primarily underpinned by PRISMA. The work is interesting atleast in 2 fronts: ii) it conducts a comprehensive literature review of impact of heatwaves on health through an innovative perspective - connecting the review to a theoretical perspective on vulnerability; ii) it brings both the dimensions of health impacts - mortality and morbidity. I believe the work makes good contribution to the literature. However, it falls short on a major front and needs major revisions to be published. I am attaching the pdf with comments throughout the text but here are my major comments - 

1. The paper starts with highlighting the vulnerability components (exposure, sensitivity and adaptive capacity), and the factors influencing those components. However, it is important that the conceptual framework of vulnerability is first discussed before highlighting the factors of vulnerability framework. There is a huge literature on this but the authors could start with IPCC vulnerability definition and also works starting with Yenneti, K., Tripathi, S., Wei, Y.D., Chen, W. and Joshi, G., 2016. The truly disadvantaged? Assessing social vulnerability to climate change in urban India. Habitat International, 56, pp.124-135.

2. The methodology section shorts fall of describing what quantitative analysis methods were used to analyse the quantitative and qualitative analysis data in the papers. How the results of both the methods were triangulated. It is also not clear how the thematic analysis of qualitative methods was used. I don't see any quotes or themes or information. The number of final articles articles (35) also seems to be too small for a systematic review. Please thoroughly revise the methodology. 

3. The results section should include tables on factors based on the vulnerability components (exposure, sensitivity and adaptive capacity) instead of characteristics based on socioeconomy, location, medical illness, etc. The arguments should also be based on the vulnerability components. At the moment all the factors are mixed up. Now this is why the vulnerability should be clearly defined in the introduction. 

4. The discussion section should also be modified as per the vulnerability definition and its components. 

5. Conclusions should highlight some implications of the study, and future opportunities for research. 

6. The paper needs thorough editing and proofreading. Too many language errors to highlight here. 

Overall, the article needs to be reframed around the vulnerability concept -  a function of exposure, sensitivity and adaptive capacity. The introduction, results and discussion need to be thoroughly reviewed and reorganised around a conceptual framework, which is currently missing. 

Thank you again and I hope the comments are useful.

Author Response

Dear Reviewer,

Thank you

Author Response

Dear Reviewer,

Thank you

Round 2

Reviewer 1 Report

1.  There are still some studies included in the review which to me does not meet the definition of heatwave due to its lack of duration included in the definition. Please review this point. Example studies: ref 51,18, 47,31,

2. The authors have still missed some studies in their review which are published in their 2012 to 2022 period. Please review this point.

Heatwaves and diabetes in Brisbane, Australia: a population-based retrospective cohort study - PubMed (nih.gov)

Joint effect of heatwaves and air quality on emergency department attendances for vulnerable population in Perth, Western Australia, 2006 to 2015 - PubMed (nih.gov)

The Value of Local Heatwave Impact Assessment: A Case-Crossover Analysis of Hospital Emergency Department Presentations in Tasmania, Australia - PubMed (nih.gov)

3. There are other systematic reviews as well which I would like the authors to review and see if they are missing any relevant studies.

Systematic review of the impact of heatwaves on health service demand in Australia - PubMed (nih.gov)

Impact of short-term exposure to extreme temperatures on diabetes mellitus morbidity and mortality? A systematic review and meta-analysis - PubMed (nih.gov)

Cardiorespiratory effects of heatwaves: A systematic review and meta-analysis of global epidemiological evidence - PubMed (nih.gov)

4.  The authors have not summarised the overall strength of evidence of vulnerability. One approach they could consider is the approach used by  Son et al 2019 (Table S1) that categorised evidence into categories of strong evidence, limited/suggestive evidence, weak evidence and no evidence.

Temperature-related mortality: a systematic review and investigation of effect modifiers - IOPscience

5. Please add a weighted bar plots indicating the percentage of the risk of bias judgements within each bias domain across reviewed studies. See an example here: Heat exposure and cardiovascular health outcomes: a systematic review and meta-analysis - The Lancet Planetary Health (Figure 5)

Reviewer 2 Report

Thank you for providing revisions to the manuscript. Some previous comments and suggestions have not been adequately addressed and the following further revisions are suggested.

1.      The manuscript needs a professional English language review. There are numerous grammatical errors throughout (previously recommended and not addressed).

2.      While the search strings have been improved, the phrases chosen are very restrictive (eg ‘ambulance callouts’, ‘renal disease’). The aim of a review paper is to capture all possible combinations, and then review towards the criteria. I would strongly suggest revision of the search string to be less restrictive. For example, globally, ambulance callouts are variously known as ambulance dispatches, ambulance events, ambulance calls, etc. Using the phrase ‘ambulance*’ covers these alternatives. Same for hospital events – these could be hospital admissions, presentations, events, etc. Similarly, by including the phrase ‘disease’ in the medical condition strings, you are potentially eliminating phrases such as ‘cardiovascular outcomes’ or ‘cardiovascular events’, etc. So using the phrase ‘cardiovascular*’, etc will capture a greater pool of papers.

3.       Line 44-45 and line 74: IPCC = Intergovernmental Panel on Climate Change, as used in Reference #4. Please update in the text (previously recommended and incorrectly addressed).

4.       Lines 64-66: The reference from 1961 is not appropriate here. Our understanding of the impacts of extreme heat on physiological pathways has developed considerably in the last 60 years. For example, mental health impacts are not listed here, yet the review finds mental health impacts (lines 323-327) as a new finding, which is not the case - there is ample evidence of the impact of extreme heat on mental health. Please update this reference to include a more recent overview of the pathways to impact of extreme heat on various physiological systems, including mental health.

5.       Lines 74-77: The gap this research attempts to fill could be better articulated. The emphasis of your research is to identify vulnerability/sensitivity factors, not provide “comprehensive evidence on heatwave impact on mortality and morbidity”. Please reword to reflect the emphasis/key differentiating point of your research as identifying vulnerability factors to heatwave impact (previously recommended, not adequately addressed).

6.       Lines 131-134: It would be of benefit to the reader to be more specific about the climate type of the region under study, rather than classifying per country. For example, Toloo 2014 concentrates on the Brisbane region in Australia, which is not classed as ‘arid to semi arid’. Please review Table 1, and the specific location under study for each of the included articles, update any climate types to the specific region of study, and update the text in lines 131-134 accordingly. This is especially relevant for studies in larger countries such as the USA, China and Australia.

7.       Line 195: Table 2, not Table 3.

8.       Line 250: Table 3, not Table 4.

9.       Section 3.1.1: Please quantify the number of studies rather than using ‘most’ or ‘several’.

10.   Lines 280-282: As presented in the manuscript, there is no evidence for the gender inconsistency being a result of “pre-existing medical conditions, social support and type of work exposure”. Furthermore, line 282-4 could be better worded as “Males may be at increased risk, as in some locations, outdoor work roles are predominantly male dominated [ref]. However, females may be at greater risk, as women have a higher surface-to-mass ratio and greater subcutaneous fat thickness compared to men [ref].” This may partly explain the inconsistencies of vulnerability by gender.

11.   Lines 287-288: Please provide a reference for the statement on low SES and health (previously recommended and not addressed).

12.   Line 274: As discussed above (#4), mental health impacts from extreme heat have been previously extensively researched and reported. This current research has not ‘discovered’ this. Please revise accordingly.

13.   Line 351-352: What does ‘greater heatwaves’ refer to? Frequency, intensity and/or duration? Please clarify and provide a reference to this statement regarding heatwaves in SE Asian countries.

14.   Lines 379-381: In Section 4.4, you state that ‘each locality setting must be examined for a better understanding’, which is a thoughtful and accurate response based on the differences in your findings in different locations. However, this statement is not correlated in the conclusion, where you state policymakers can use these findings to reduce the impact of heatwaves. This statement implies a generalisability that is not present in your findings. It would be helpful to include a statement on generalisability of your findings in the discussion, and then draw on this in your conclusion (previously recommended and not addressed).

Reviewer 3 Report

Dear authors, 

Thank you for revising the manuscript. Some significant efforts were made to edit and revise the manuscript. However, I still have a couple of major concerns with the manuscript:

1. It is not clear why the authors have highlighted 'vulnerability factors (Sensitivity)' in sections 3.1.2, 3.2.2 and 4.2 but have discussed exposure and adaptive capacity along with the sensitivity in the introduction. Vulnerability is a function of all 3 (exposure, sensitivity and adaptive capacity) and hence attention to all 3 is important. I guess attention to section 2.4, where the authors discussed vulnerability (sensitivity) factors may partially help address this issue. 

2. The manuscript still needs significant proofreading and editing - several sentences are still very weak in grammar and language. 

Thanks again.